# The Overexpression of Peanut (*Arachis hypogaea* L.) *AhALDH2B6* in Soybean Enhances Cold Resistance

**DOI:** 10.3390/plants12162928

**Published:** 2023-08-12

**Authors:** Mingyu Yang, Yuhan Teng, Tong Yue, Ziye Wang, Guanghui Feng, Jingwen Ruan, Shi Yan, Yuhong Zheng, Ling Zhang, Qingshan Chen, Fanli Meng

**Affiliations:** 1College of Agriculture, Northeast Agricultural University, Harbin 150030, China; yangmingyu0723@outlook.com (M.Y.); tengyuhannbplus@163.com (Y.T.); 18903659093@163.com (T.Y.); wangziye0925@163.com (Z.W.); fengguanghui0714@163.com (G.F.); ruanjw4743@163.com (J.R.); 773766318wangzha@gmail.com (S.Y.); 2Northeast Institute of Geography, Agroecology Chinese Academy of Sciences, Harbin 150081, China; 3Jilin Academy of Agricultural Sciences, Changchun 130033, China; zhengyuhong520@163.com

**Keywords:** soybean, transgenic breeding, ALDH, cold stress, RNA-seq

## Abstract

Soybeans are the main source of oils and protein for humans and animals; however, cold stress jeopardizes their growth and limits the soybean planting area. Aldehyde dehydrogenases (ALDH) are conserved enzymes that catalyze aldehyde oxidation for detoxification in response to stress. Additionally, transgenic breeding is an efficient method for producing stress-resistant germplasms. In this study, the peanut *ALDH* gene *AhALDH2B6* was heterologously expressed in soybean, and its function was tested. We performed RNA-seq using transgenic and wild-type soybeans with and without cold treatment to investigate the potential mechanism. Transgenic soybeans developed stronger cold tolerance, with longer roots and taller stems than P3 soybeans. Biochemically, the transgenic soybeans exhibited a decrease in malondialdehyde activity and an increase in peroxidase and catalase content, both of which are indicative of stress alleviation. They also possessed higher levels of ALDH enzyme activity. Two phenylpropanoid-related pathways were specifically enriched in up-regulated differentially expressed genes (DEGs), including the phenylpropanoid metabolic process and phenylpropanoid biosynthetic process. Our findings suggest that *AhALDH2B6* specifically up-regulates genes involved in oxidoreductase-related functions such as peroxidase, oxidoreductase, monooxygenase, and antioxidant activity, which is partially consistent with our biochemical data. These findings established the function of AhALDH2B6, especially its role in cold stress processes, and provided a foundation for molecular plant breeding, especially plant-stress-resistance breeding.

## 1. Introduction

Soybean (Glycine max) is an important cash crop that provides humans with edible oils and proteins. Furthermore, by fixing atmospheric nitrogen, soybeans can serve as a great protein source for animal feed and improve soil fertility [1]. Soybean is a thermophilic crop that is cultivated widely [2], with rising demand having led to rapidly increased planting areas in cold regions, such as northeast China, in recent years. However, global climate change is increasing the frequency of extreme climate events, such as drought and low temperatures [3]. Low temperature can lead to changes in cell membrane function, resulting in an imbalance of water metabolism and reduced photosynthesis. This can cause plants to wither or even die, leading to a decrease in crop yield [4,5]. Soybean is susceptible to low temperatures, which cause soybean ripening to be delayed, resulting in shorter plants and smaller leaves, reducing the number of soybean pods and ultimately resulting in a lower soybean yield [6]. Low temperatures may exert negative effects on soybeans at all developmental stages by interfering with membrane integrity in the seed imbibition process, preventing the seed from germinating and emerging, damaging the green stems and leaves affecting the early survival and growth of soybean seedlings [2,7,8], delaying flowering and seed abortion in the pod [9,10], and shrinking seed size and delaying maturity in the reproductive stage [10]. Identifying genes responsible for low-temperature tolerance to mitigate low-temperature-mediated damage is critical for long-term soybean breeding.

Under low temperatures, chloroplasts overproduce reactive oxygen species (ROS), leading to excessive ROS accumulation inside the cell [11]. High levels of ROS are often accompanied by increased membrane lipid peroxidation [12]. Malondialdehyde (MDA), a three-carbon aldehyde, is a toxic byproduct of lipid oxidation. Owing to their chemical reactivity and toxicity, endogenous aldehydes interfere with cellular metabolism, causing DNA damage and cell death [13,14,15]. Under low temperatures, crops have evolved mechanisms to improve their tolerance to low temperatures [16]. Peanut is an important oil and cash crop, which can maintain normal plant development by removing excess aldehydes through aldehyde dehydrogenase (ALDH) [17]. ALDH is an aldehyde scavenger that can protect plant membranes from stress-induced ROS damage [14,15]. ALDH uses NAD+ or NADP+ as cofactors to catalyze the oxidation of a wide range of endogenous and exogenous aldehydes to their corresponding non-toxic carboxylic acids, generating NADH or NADPH to maintain redox homeostasis [14,15]. Moreover, ALDH is involved in secondary metabolism, especially in the synthesis of amino and retinoic acids and osmoprotectant production, such as glycine betaine [18,19,20].

We have previously described and annotated the entire *ALDH* gene superfamily in *Arachis hypogaea* Linn [17]. However, the function and application of the *ALDH2* family in preventing plant damage under low-temperature stress requires further investigation. In this study, the expression pattern of the *ALDH2* family at lower temperatures was analyzed in *Arachis hypogaea* in cold-tolerant TF15 and cold-sensitive FH25 varieties using RT-qPCR. We then determined the function of the peanut *ALDH* gene *AhALDH2B6* in preventing plant damage due to low-temperature stress in soybean by heterologous expression. Finally, we performed RNA-seq to elucidate the regulatory mechanism of *AhALDH2B6*. The aim of this study was to breed new soybean cultivars with tolerance for low temperatures.

## 2. Results

### 2.1. Expression of AhALDH2 Genes in Peanut Plants under Low-Temperature Stress

Based on our previous findings [17], we selected 20 *AhALDH2* genes for further investigation. To determine the expression pattern of *AhALDH2* family genes under lower temperature treatment, one cold-tolerant variety, TF15, and cold-sensitive variety, FH25, peanut plants were stored at 6 °C in a chamber for 9 d (Figure 1A). Under 9 d of continuous low-temperature stress, *AhALDH2B2*, *AhALDH2B3*, *AhALDH2B4*, *AhALDH2B6*, *AhALDH2B7*, *AhALDH2C1*, and *AhALDH2C3* exhibited significant up-regulation in the roots of both peanut cultivars (Figure 1B). Furthermore, ALDH enzyme activity in TF15 was higher compared to that in FH15 (Figure 1C).

To further confirm the function of *ALDH2* genes in low-temperature tolerance, we employed a yeast expression vector [21]. The up-regulated genes were cloned into the pYES2 vector and transformed into yeast. An empty pYES2 vector was used as a control. There were no significant differences between all expression vectors and the empty vector yeast under normal culture conditions (Appendix A). Under low temperatures, the survival rate of yeast cells expressing *AhALDH2B6* was significantly higher than that of the control and other yeast cells (Appendix A). Additionally, the expression level of *AhALDH2B6* in TF15 cells was significantly higher than that of FH15. To further characterize *AhALDH2B6*, we also used qPCR to examine its expression level in different tissues. We found that *AhALDH2B6* was mainly expressed in the roots, stems, nodules, and flowers, followed by the leaves and seeds, and it is rarely expressed in pods (Figure 1D).

### 2.2. Soybean Transformation and Positive Line Identification

To explore the stress resistance of *AhALDH2B6* further, we overexpressed *AhALDH2B6* in soybean via *Agrobacterium*-mediated transformation. The *AhALDH2B6* coding sequence was driven by the 35S promoter (Figure 2A). Fifteen transgenic soybean plants were obtained in T0, all of which were confirmed by PCR to have the expected size of the selection marker, *bar* gene (422 bp), and target gene, *AhALDH2B6* (1272 bp) (Figure 2B,C). After three generations of propagation, three homozygous overexpression lines were obtained, named line 1-1, line 2-2, and line 4-1. After 9 d of low-temperature treatment, *AhALDH2B6* was found to be highly expressed in lines 1-1, 2-2, and 4-1 (Figure 2D).

### 2.3. AhALDH2B6 Increases Soybean Low-Temperature Tolerance

To evaluate *AhALDH2B6*’s function in soybean, we separately assessed the low-temperature tolerance in transgenic and P3 soybean during the germination and emergence stages. After cold-temperature treatment, the transgenic soybean root was significantly longer than that of the P3 plants (Figure 3A,B). The height of transgenic plants ranged from 10.40 ± 0.91 cm to 11.27 ± 1.366 cm, significantly higher than that of P3 plants (9.50 ± 1.127 cm) (Figure 3C,D).

### 2.4. Overexpression of AhALDH2B6 Reduced Lipid Peroxidation and Increased Protective Enzyme Activity

MDA content reflects the degree of lipid peroxidation damage in plant membranes. The ALDH enzyme activity and MDA content of transgenic lines and P3 were measured under regular growth and low-temperature conditions to see how *AhALDH2B6* overexpression affects soybean biochemically. Under normal growth conditions, ALDH enzyme activity and MDA content did not differ between transgenic lines and P3 (Figure 4A,B). Under low-temperature stress, ALDH enzyme activity increased in transgenic soybean and P3 plants with a much higher elevation in transgenic soybean. Moreover, low temperatures led to an increase in MDA content in P3 plants, which was higher than in transgenic lines (Figure 4B). Plants accumulate protective enzymes, such as POD and CAT, in response to stress, which can lower cellular ROS levels. Figure 4C,D show that transgenic soybean had higher protective enzyme activity than P3 under low-temperature stress.

### 2.5. RNA-Seq Analysis of Transgenic Lines and P3 Soybeans

After a low-temperature treatment for 9 days, radicles from transgenic soybean and recipient soybean P3 were sampled and, respectively, named B9 and B11 hereafter. As a control, the same plants were mock-treated at 25 °C and were named C2 and C5, respectively. Two RNA libraries were constructed per sample as biological replications. Clean sequencing data for each sample were compared to the Williams 82 reference genome. We detected 2850 DEGs in transgenic soybean treated under cold stress, compared to growth at 25 °C, including 1533 up-regulated and 1317 down-regulated DEGs (Figure 5A). We found 1624 DEGs in P3 soybean after cold treatment compared to plants grown at 25 °C, including 498 up-regulated and 1126 down-regulated DEGs (Figure 5B). The transgenic and P3 soybeans shared 58 up-regulated DEGs, while transgenic soybean had 1480 specific DEGs, and P3 plants had 445 specific DEGs (Figure 5C, Appendix A). Transgenic and P3 soybean shared 87 down-regulated DEGs, while transgenic soybean had 1230 specific DEGs, and P3 plants had 1039 specific DEGs (Figure 5D, Appendix A).

### 2.6. Gene Ontology and KEGG Enrichment Analysis of Differentially Expressed Genes

We performed a GO analysis of DEGs to investigate their functions, focusing on the specific 1480 up-regulated and 1230 down-regulated genes in transgenic soybean. Significant GO terms were enriched in response to stimulus (GO:0050896), response to chemical (GO:0042221) for 1480 up-regulated DEGs in terms of biological process (BP), followed by response to organic substance (GO:0010033), response to endogenous stimulus (GO:0009719) and response to oxygen-containing compound (GO:1901700) (Figure 6A). For molecular function, significant GO terms were enriched in oxidoreductase activity (GO:0016491) and transcription regulator activity (GO:0140110) (Figure 6B).

For down-regulated DEGs in biological processes, significant GO terms were enriched in response to stimulus (GO:0050896), response to chemical (GO:0042221), and response to biotic stimulus (GO: GO:0002831) (Figure 6C). In addition, for molecular function, significant GO terms were enriched for transcription regulator activity (GO:0140110), DNA-binding transcription regulator activity (GO:0003700), and sequence-specific DNA binding (GO:0043565) (Figure 6D). Notably, two phenylpropanoid-related pathways were specifically enriched in up-regulated DEGs, including the phenylpropanoid metabolic process and phenylpropanoid biosynthetic process (Figure 6D). For molecular function, multiple oxidoreductase-related functions were specifically found in up-regulated DEGs, including peroxidase, oxidoreductase, monooxygenase, and antioxidant activity.

KEGG analysis was performed on the specifically up-regulated and down-regulated DEGs to further analyze the pathways involved in improving cold tolerance in transgenic soybean. The up-regulated DEGs were enriched in the biosynthesis of secondary metabolites (gmx01110), followed by plant hormone signal transduction (gmx04075) and phenylpropanoid biosynthesis (gmx00940) (Figure 6E). The down-regulated genes were enriched in plant hormone signal transduction (gmx04075), starch and sucrose metabolism (gmx00500), and protein processing in endoplasmic reticulum (gmx04141) (Figure 6F). Of note, the plant–pathogen interaction was found in down-regulated DEGs, which is consistent with our GO analysis response to biotic stimulus (GO: GO:0002831).

## 3. Discussion

The importance of the *ALDH2* gene family in multiple plant species and under stress conditions has been studied [17]. The essential function of *ALDH2* is useful for various applications and transgenic breeding and has become popular in recent years. Many plant studies have revealed that ALDH upregulation and ROS reduction are common features of the activation of stress response pathways [22]. In this study, we heterologously overexpressed a peanut *ALDH* gene, *AhALDH2B6*, in soybeans. Our transcriptome findings suggest that genes enriched in response to abiotic stimuli were up-regulated, which partially helps explain the increased cold tolerance of transgenic soybean. The research indicates that the up-regulated genes are mainly involved in oxidoreductase-related function and phenylpropanoid metabolic process, while the down-regulated genes are associated with starch and sucrose metabolism as well as plant hormone signal transduction.

In this study, *AhALDH2B6* overexpression resulted in less MDA, which improved cold tolerance. *OsALDH2a*, another rice *ALDH* orthologous gene, was found to be involved in submerged conditions [23]. An anaerobic environment strongly induces its mRNA levels. A higher concentration of ALDH could partially explain why rice plant has a stronger submergence tolerance than other plant species. *ALDH* gene stress responses have been investigated in other plant species. *ALDH2C4* mutation in Nicotiana benthamiana and Solanum tuberosum caused susceptibility to low temperatures and accumulated ROS and MDA [24]. Plant drought stress tolerance was conferred by ectopic expression of the Arabidopsis gene *AtALDH2B7* [25]. The consistent expression of maize *ALDH22A1* in tobacco led to improved stress tolerance and reduced MDA [26].

Similarly, other studies have reported similar results. For example, one proteomic analysis in *Citrus junos* revealed that the proteins in starch and sucrose metabolism, secondary metabolites biosynthesis, and phenylpropanoid biosynthesis are differentially abundant, which is consistent with our research [27]. Another de novo transcriptome sequencing in *Elymus nutans* found that cold stress is related to secondary metabolism pathways, which is consistent with the up-regulated pathways in this study [28]. The specific up-regulated genes which were involved in oxidoreductase-related function have also been reported by others. Oxidoreductase activity and transcription regulator activity pathway genes were found to be differently expressed in Alfalfa by RNA-seq analysis in response to cold stress [29]. Another study in rice found that a higher number of genes involved in oxidoreductase activity might contribute to cold tolerance [30]. Moreover, the identification of DEGs in Pisum sativum using RNA-seq analyses confirmed that oxidoreductase-related genes were enriched after cold treatment [29]. Notably, our result that the down-regulated genes were involved in the plant–pathogen interaction pathway supports the notion that there exists balancing trade-offs between biotic and abiotic stress responses [31].

In conclusion, in the present study, we showed that transgenic soybeans expressing peanut *AhALDH2B6* had greater cold tolerance, with longer roots and taller stems, and a lower content of MDA and higher activities of POD, CAT, and ALDH compared to P3 plants. RNA-seq data showed that *AhALDH2B6* altered the expression of genes involved in oxidoreductase-related functions such as peroxidase, oxidoreductase, monooxygenase, and antioxidant activity. These findings shed light on the function of AhALDH2B6 and its role in abiotic stress processes and highlight the potential of transgenic breeding in facilitating the growth of soybean in cold climates.

## 4. Materials and Methods

### 4.1. Plant Materials and Growth Conditions

Transgenic soybean and recipient soybean P3 were grown at 25 °C in a greenhouse under LDs (16 h/8 h light/dark). Transgenic and P3 soybeans were provided by the Institute of Agricultural Biotechnology, Jilin Academy of Agricultural Sciences, Changchun, China. Tolerant (TF15) and sensitive (FH25) peanuts were incubated at 25 °C in a greenhouse under LDs (16 h/8 h light/dark. Relative humidity: 50%. The light intensity was 300 μmolm^−2^s^−1^).

### 4.2. Vector Construction

Total RNA was extracted from the roots of three-week-old peanuts and reverse-transcribed into cDNA. The *AhALDH2* genes from peanut were amplified using corresponding primers (Appendix A). The pCAMBIA3301 vector was digested with *Nco I* and *BstE II*, while *Kpn I* and *EcoR I* were utilized for the digestion of the pYES2 vector. The *AhALDH2B6* gene was then cloned into the pCAMBIA3301 (named pCAMBI3301-AhALDH2B6) vector, while those of *AhALDH2B2*, *AhALDH2B4*, *AhALDH2B6*, *AhALDH2B7,* and *AhALDH2C1* gene coding sequence were cloned into yeast expression vector pYES2 containing GAL1 promoter and URA3 as selective markers.

### 4.3. Genetic Transformation and Progeny Identification in Soybean

The pCAMBI3301-AhALDH2B6 recombinant plasmid was transformed into Agrobacterium EHA101. Regenerated plants expressing pCAMBI3301-AhALDH2B6 were obtained, as previously described [32,33]. Genomic DNA was extracted from transgenic progeny, and positive plants were identified through PCR using corresponding primers (Appendix A).

### 4.4. Yeast Transformation and Low-Temperature Treatment

The yeast protein expression under low-temperature treatment was analyzed as previously published [21]. Briefly, the recombinant vector and pYES2 vector were transformed into INVSCI-competent cells using a yeast transformation kit, and the transformation products were plated on SD/-Ura (2% (*w*/*v*) glucose) plates and cultured at 30 °C for 3–5 days to screen for positive clones. The positive clones were subsequently resuspended in sterile water, plated onto SG/-Ura (2% (*w*/*v*) galactose) medium, and incubated at 30 °C for 3–5 days to screen positive clones. The positive clones were inoculated in 15 mL SD/-Ura liquid medium and cultured at 30 °C until the OD_600_ value reached 0.4. The yeast culture was centrifuged at 8500 rpm for 1 min. The cells were resuspended in 1–2 mL SG/-Ura medium and then cultured in a 5 mL induction medium for 24 h at 30 °C. The yeast was treated at −20 °C for 3, 6, 9, 12, or 24 h, followed by a recovery period of 9 h at 30 °C, at the end of which cell density (OD_600_) was measured.

### 4.5. Low-Temperature Stress Treatment

Transgenic soybean and P3 seeds in the same growth stage were selected. The seeds were disinfected with 70% (*v*/*w*) ethanol for 1 min and then soaked in 5% (*w*/*v*) sodium hypochlorite solution for 15 min before being rinsed 5–6 times with sterile water. Then, the completely sterilized soybean seeds were soaked in 25 °C sterile water for 12 h, followed by incubation in a dark growth chamber for 24 h. For the germination stage, three transgenic strains and P3 were planted in vermiculite for two weeks and then transferred into a lower temperature chamber of 6 °C. Plant height was measured after 12 d, and root length was measured after 9 d. TF15 and FH25 peanuts were sown in sand and then transferred to a greenhouse at 25 °C under LDs (16 h/8 h, light/dark). Three weeks later, TF15 and FH25 peanut plants were stored in a 6 °C chamber for 9 d (Figure 1A). We used a commercial kit (Michy Bio, Suzhou, China) to determine physiological indicators. ALDH (Cat. #M0608A) enzyme activity was measured in peanut. Physiological indices were measured in transgenic soybeans, including ALDH (Cat. #M0608A) and MDA (Cat. #M0106A), POD (Cat. #M0105A), and CAT (Cat. #M0104A) activities after 12 d of low-temperature stress treatment, according to the manufacturer’s instructions.

### 4.6. RNA Library Construction

We performed RNA-seq analysis using line 1-1 and P3 as a control. The soybean was treated with/without cold treatment for 9 d. The radicles were sampled for RNA extraction using an RNA extraction kit (Kangwei Century Biotechnology, Beijing, China). Berry Genomics Company performed library construction and transcriptome sequencing.

### 4.7. qRT-PCR Validation of Gene Expression

For peanut, to investigate the expression pattern of *AhALDH2* family genes, after 9 d of low-temperature treatment, TF15 and FH25 roots were collected for qPCR verification. Meanwhile, the expression pattern of *ALDH2* expression in different tissues of peanut was analyzed. For soybeans, after 9 d of low-temperature treatment, three T3 generation transgenic homozygous soybeans (line 1-1, line 2-2, line 4-1) and P3 soybean radicles were used for qPCR verification. Total RNA was extracted from plants using TRIzol reagent. The reverse transcription kit was used for cDNA synthesis and the removal of genomic DNA. The cDNA sequence was used as a template to analyze differences in gene expression using the SYBR qPCR Master Mix kit (vazyme, Nanjing, China). The LightCycler 480 II instrument was used for qPCR analysis. The 2^−ΔΔCT^ method was used to calculate relative expression. *GmActin6* and *Arachis*-*Actin11* were used as internal reference genes. Appendix A shows the list of the primers used for qPCR.

### 4.8. Quality Control and Reads Mapping

The quality of the RNA-seq raw reads was analyzed using Fast QC software (version 0.11.9) [34], which was used to check and filter the adaptors and low-quality bases using the Trim galore software (version 0.6.7). Using the Hisat2 software, the clean reads were mapped to the soybean reference genome (Glycine_max_v4.0, NCBI) [35]. The mapped reads were then quantified using the feature Counts software (version 2.0.1) [36]. DESeq2 (version 1.36.0) [37] was used to normalize quantified reads and perform differential expression analysis between sample groups to identify differentially expressed genes between transgenic and wild-type at room temperature versus cold treatment and to prepare data for functional enrichment analysis. The raw data from this experiment were submitted to the NCBI SRA database (ID: PRJNA932570).

### 4.9. Gene Ontology and KEGG Pathway Analysis

The R package AnnotationHub (version 3.6.0) was used to obtain the annotation information files for soybean in the database. Using the R package ClusterProfiler (v4.6.0), down-regulated and up-regulated differentially expressed genes (DEGs) lists were analyzed for gene ontology (GO) gene function clustering and enrichment. Using the R package ClusterProfiler (v4.6.0), KEGG pathway annotation analysis of differentially expressed genes was performed, the metabolic pathways of gene products in cells were systematically analyzed, and the function of these gene products was determined.

### 4.10. Data Analysis

In this study, SPSS software was used for data analysis, and one-way ANOVA and *t*-test were used for the significance test.

## Figures and Tables

**Figure 1 plants-12-02928-f001:**
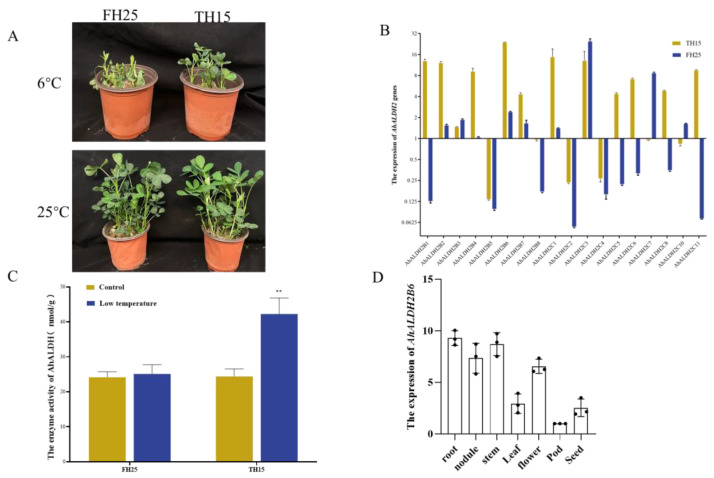
Analysis of the expression pattern of the *ALDH2* gene family and measurement of ALDH enzyme activity. (**A**) Phenotypes of FH15 and TH25 peanuts at low and normal temperatures. (**B**) Relative expression levels of the *ALDH2* gene family in FH15 and TH25 peanuts treated at low temperatures (<1 represents down-regulated gene expression, and >1 represents up-regulated gene expression). (**C**) ALDH enzyme activity of FH15 and TH25 peanuts at low and normal temperatures. ** *p* < 0.01, *t* test. (**D**) Expression pattern of *ALDH2* gene under low-temperature treatment.

**Figure 2 plants-12-02928-f002:**
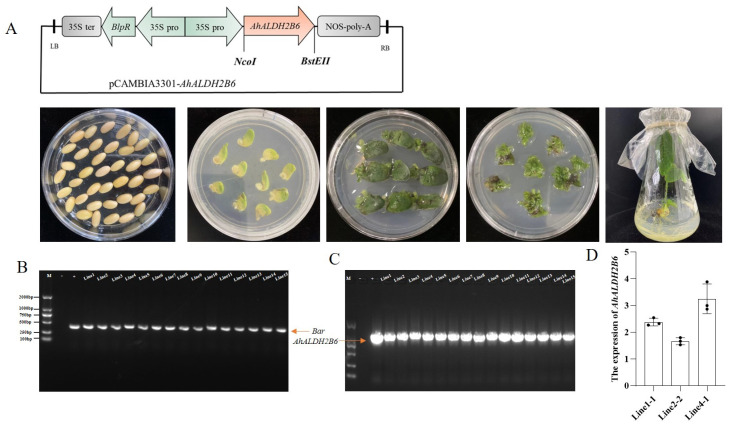
*AhALDH2B6* overexpressed in soybean plants. (**A**) Process of genetic transformation of soybean with pCAMBIA3301-AhALDH2B6 vector. (**B**,**C**) T0 Positive transgenic soybean was verified using PCR: bar 422 bp, *AhALDH2B6* 1272 bp. (**D**) Expression levels of *AhALDH2B6* transgenic soybean between different lines at low temperatures.

**Figure 3 plants-12-02928-f003:**
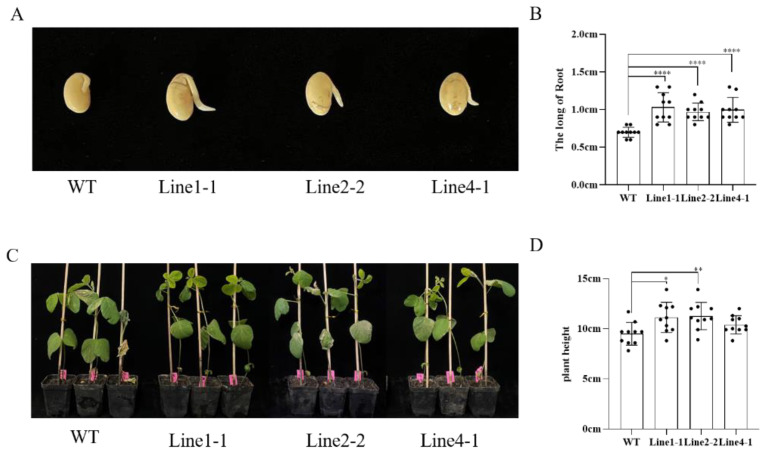
Low-temperature stress phenotypes of transgenic soybean. (**A**,**B**) Phenotype and root length of transgenic soybean and P3 under low-temperature treatment at the germination stage. (**C**,**D**) Phenotype and plant height of transgenic soybean and P3 under low-temperature treatment at the emergence stage * *p* < 0.05, ** *p* < 0.01, **** *p* < 0.0001.

**Figure 4 plants-12-02928-f004:**
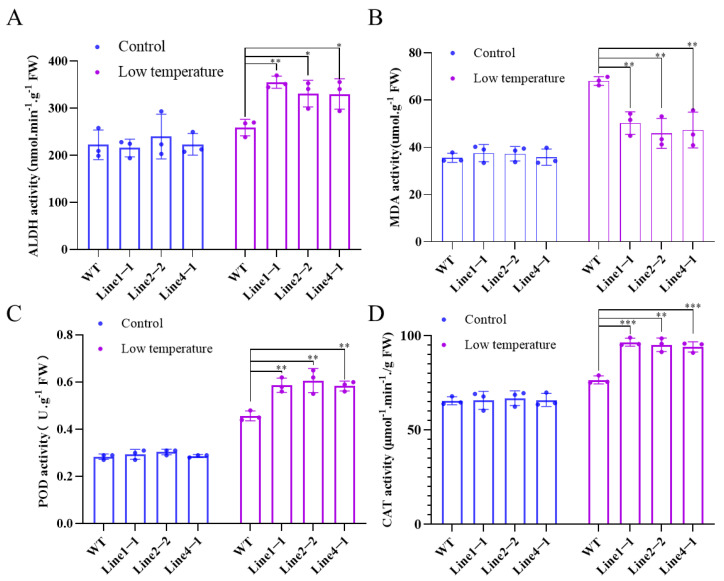
Determination of physiological parameters of transgenic soybean and P3 under low-temperature stress. (**A**). ALDH activities (**B**) MDA activities. (**C**) POD activities. (**D**) CAT activities * *p* < 0.05, ** *p* < 0.01, *** *p* < 0.0001.

**Figure 5 plants-12-02928-f005:**
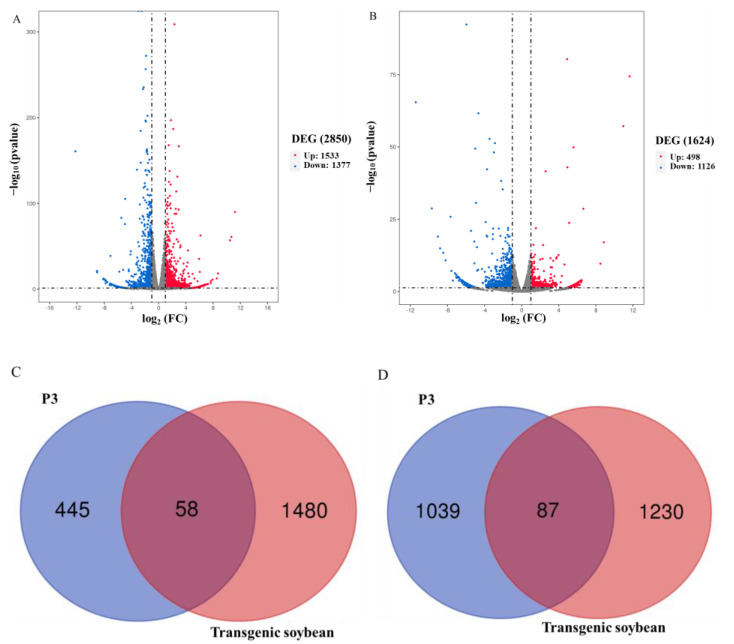
Comparison of differentially expression genes (DEGs) between transgenic lines and P3 under low and normal growth temperatures. (**A**) DEGs analyzed in transgenic soybeans under cold stress compared to 25 °C growth conditions. (**B**) DEGs analyzed in P3 soybeans under cold stress compared to 25 °C growth conditions. (**C**) Up-regulated DEGs in transgenic and P3 soybeans. (**D**) Down-regulated DEGs in transgenic and P3 soybeans.

**Figure 6 plants-12-02928-f006:**
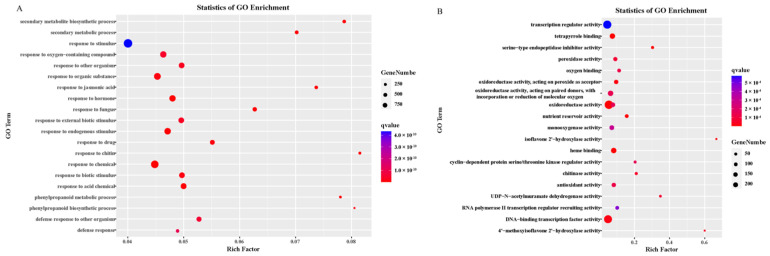
Gene Ontology (GO) and KEGG enrichment analysis of differentially expressed genes (DEGs). The color from red to blue represents the value from low to high. (**A**–**D**) In terms of BP and MF, GO enrichment of 1480 up-regulated DEGs and 1230 down-regulated DEGs. (**E**) KEGG analysis on up-regulated. (**F**) KEGG analysis on down-regulated.

## Data Availability

The RNA-seq data in this study have been uploaded into the NCBI database; the accession number is PRJNA932570. https://dataview.ncbi.nlm.nih.gov/object/PRJNA932570?reviewer=8rrkdsernf9lut66jgfhl32o3j (accessed on 8 February 2023).

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
