# Peer review of "The Overexpression of Peanut (Arachis hypogaea L.) AhALDH2B6 in Soybean Enhances Cold Resistance"

_plants, 2023, doi:10.3390/plants12162928_

Round 1

Reviewer 1 Report

The manuscript titled "Overexpression of peanut... enhances cold resistance" by Yang et al. has focused on establishing the functional role of AhALDH2B6 in cold stress tolerance by overexpressing it in soyabean. The findings in this study might have significant applications in developing abiotic stress-tolerant crops. Overall, the manuscript is well-written and informative. However, to further improve its quality, the authors should consider addressing the following points: 

1. In the abstract, line no. 11, what is the meaning of "many species"? Authors are suggested to rewrite the sentence for better clarity for the readers. 

2. Is there a particular justification for taking this ALDH gene only from peanuts? Has this gene's availability been checked for over-expression in soybeans by the authors? 

3. It is advised that authors include a table listing the top 20 genes that were up- and down-regulated based on their RNA Seq data. 

4. Authors are suggested to provide a better-quality image in Figure 6. 

5. Include a suitable reference for the method mentioned in lines 307–308.

 Minor editing of English language required

Reviewer 2 Report

Title: Overexpression of peanut (Arachis hypogaea L.) AhALDH2B6 2 in soybean enhances cold resistance

Plants under cultivation are exposed individually or simultaneous to different abiotic stress. This possess challenges to the plant research scientist. In the present study transgenic soybeans overexpressing peanut AhALDH2B6 showed greater cold tolerance, with longer roots and taller stems, and a lower content of reactive species scavenging physiological and biochemical parameters viz., MDA, higher activities of POD, CAT, and ALDH compared to the control soybean P3 plants. This study also further confirmed the AhALDH2B6 altered the expression of genes involved in oxidoreductase related function such as peroxidase, oxidoreductase, monooxygenase and antioxidant activity through RNA-seq data analysis. These findings clearly shows the role and functions of AhALDH2B6 under in abiotic stress conditions and also highlighted the potential of transgenic breeding to mitigate soybean in cold climates.

Overall, manuscript content is interest to the readers and highlights the foundation for molecular plant breeding, especially plant stress resistance breeding. However, I have some suggestions to improve the presentation of this manuscript. 

Abstract:

Line 11, Soybean is an important cash crop.

Line 17, Transgenic soybeans developed

Results:

Line 64 & 74, Whether Arachis hypogaea in cold-tolerant (Tiefeng15) is similar to TF15 and cold-sensitive (Jinhua15) is similar to FH25?. If so, these must be consistent in last paragraph of introduction and first section 2.1 of Results.  

Line 91-92, Variety identity number is not correct in Fig 1A, 1B. 1C and 1D and also title of the Fig. 1. Fig 1A and Fig 1C, tolerant and sensitive variety data presentation order must be same for easy comparison and comprehension. For example, Fig 1A, date of sensitive variety is given in first column whereas in Fig 1C, date of sensitive variety is given in second column.

Line 96-97, gene expression).    

Line 112, (D) Expression levels of AhALDH2B6 transgenic

Line 118, Delete word, After

Line 191-194, Clarity of the figures requires improvement. It would be better split these figures and present it in two figures for more clarity.

Discussion:

Line 200-201, Cite reference

Line 202-204, Cite reference

Line 206, genes enriched in response to abiotic stimuli were upregulated,

Materials and methods:

Line 251, include the relative humidity and light intensity used

Line 295, Any reason radical was used for RNA seq instead of plumule?

Reviewer 3 Report

The approach of the authors for the development and relevance of the subject is adequate, additionally the information provided is orderly and coherent, however i have some  remarks:

1-     The introductory part requires bibliographic updating. Authors described the importance of soybean and the role of aldehyde in plant response to cold stresss. The manuscrit talks about the isolation of a gene from Arachis hypogaea L. However, authors did not talk about the importance of Arachis hypogaea L. plants as well as the  effect of cold stress on plants. I strongly suggest to add some sentences in the introduction so the tex twill be easy to read.

 2-     Line 74 :  authors used two varieties of peanut plants presenting contrasting behaviour toward cold stress :  a tolerante one TF15 and a cold sensitive variety FH25. Then authors talk about a third variety : JH15. Please clarify the behaviour of this variety. why choosing it ? or is it a mistake (i guess)?

3-     ALDH enzyme activity in TF15 was higher compared to that in JH15 (Figure 1C). However, authors did not describe the mesurement of enzyme activity in Materiel and methods section. Please add dit

 4-     AhALDH2B2, AhALDH2B3, AhALDH2B4, AhALDH2B6, 76 AhALDH2B7, AhALDH2C1, AhALDH2C3 genes exhibited significant up-regulation in both peanut varieties. In which tissue ? which organ ? please add the information

 5-     Line 222 : Citrus junos must appear in italic. The same for Elymus nutans line 225
